# Graph Cuts with Arbitrary Size Constraints Through Optimal Transport

**Chakib Fettal**                                                    *chakib.fettal@etu.u-paris.fr*
*CDC Informatique*
*Centre Borelli, UMR 9010, Université Paris Cité*

**Lazhar Labiod**                                                    *lazhar.labiod@u-paris.fr*
*Centre Borelli, UMR 9010, Université Paris Cité*

**Mohamed Nadif**                                                    *mohamed.nadif@u-paris.fr*
*Centre Borelli, UMR 9010, Université Paris Cité*

**Reviewed on OpenReview:** *https://openreview.net/forum?id=UG7rtrsuaT*

## Abstract

A common way of partitioning graphs is through minimum cuts. One drawback of classical minimum cut methods is that they tend to produce small groups, which is why more balanced variants such as normalized and ratio cuts have seen more success. However, we believe that with these variants, the balance constraints can be too restrictive for some applications like for clustering of imbalanced datasets, while not being restrictive enough for when searching for perfectly balanced partitions. Here, we propose a new graph cut algorithm for partitioning graphs under arbitrary size constraints. We formulate the graph cut problem as a Gromov-Wasserstein with a concave regularizer problem. We then propose to solve it using an accelerated proximal GD algorithm which guarantees global convergence to a critical point, results in sparse solutions and only incurs an additional ratio of $\mathcal{O}(\log(n))$ compared to the classical spectral clustering algorithm but was seen to be more efficient.

## 1 Introduction

Clustering is an important task in the field of unsupervised machine learning. For example, in the context of computer vision, image clustering consists in grouping images into clusters such that the images within the same clusters are similar to each other, while those in different clusters are dissimilar. Applications are diverse and wide ranging, including, for example, content-based image retrieval (Bhunia et al., 2020), image annotation (Cheng et al., 2018; Cai et al., 2013), and image indexing (Cao et al., 2013). A popular way of clustering an image dataset is through creating a graph from input images and partitioning it using techniques such as spectral clustering which solves the minimum cut (`min-cut`) problem. This is notably the case in subspace clustering where a self-representation matrix is learned according to the subspaces in which images lie and a graph is built from this matrix (Lu et al., 2012; Elhamifar & Vidal, 2013; Cai et al., 2022; Ji et al., 2017; Zhou et al., 2018).

However, in practice, algorithms associated with the `min-cut` problem suffer from the formation of some small groups which leads to bad performance. As a result, other versions of `min-cut` were proposed that take into account the size of the resulting groups, in order to make resulting partitions more balanced. This notion of size is variable, for example, in the Normalized Cut (`ncut`) (Shi & Malik, 2000), size refers to the total volume of a cluster, while in the Ratio Cut (`rcut`) problem (Hagen & Kahng, 1992), it refers to the cardinality of a cluster. A common method for solving the `ncut` and `rcut` problems is the spectral clustering approach (Von Luxburg, 2007; Ng et al., 2001) which is popular due to often showing good empirical performance and being somewhat efficient.

However, there are some weaknesses that apply to the spectral clustering algorithms and to most approaches tackling the `ncut` and `rcut` problems. A first one is that, for some applications, the cluster balancing is not strict enough, meaning that even if we include the size regularization into the `min-cut` problem, the groups are still not necessarily of similar size, which is why several truly balanced clustering algorithm have been proposed in the literature (Chen et al., 2017; Li et al., 2018; Chen et al., 2019). Another problem is that the balance constraint is too restrictive for many real world datasets, for example, a recent trend in computer vision is to propose approaches dealing with long-tailed datasets which are datasets that contain head classes that represent most of the overall dataset and then have tail classes that represent a small fraction of the overall dataset (Xu et al., 2022; Zhu et al., 2014). Some approaches propose integrating generic size constraints to the objective like in Genevay et al. (2019); Höppner & Klawonn (2008); Zhu et al. (2010), however these approaches directly deal with euclidean data instead of graphs.

In this paper, we propose a novel framework that can incorporate generic size constraints in a strict manner into `min-cut` problem using Optimal Transport. We sum up our contributions in this work as follows:

- We introduce a GW problem with a concave regularizer and frame it as a graph cuts problem with an arbitrarily defined notion of size instead of specifically the volume or cardinality as is traditionally done in spectral clustering.

- We then propose a new way to solve this constrained graph cut problem using a nonconvex accelerated proximal gradient scheme which guarantees global convergence to a critical point for specific step sizes.

- Comprehensive experiments on real-life graphs and graphs built from image datasets using subspace clustering are performed. Results showcase the effectiveness of the proposed method in terms of obtaining the desired cluster sizes, clustering performance and computational efficiency. For reproducibility purposes, we release our code[1].

The rest of this paper is organized as follows: Preliminaries are presented in Section 2. Some related work is discussed in section 3. The `OT-cut` problem and algorithm along with their analysis and links to prior research are given in section 4. We present experimental results and analysis in section 5. Conclusions are then given in section 6.

## 2 Related Work

Our work is related with balanced clustering, as the latter is a special case of it, as well as with the more generic problem of constrained clustering, and GW based graph partitioning.

### 2.1 Balanced Clustering

A common class of constrained clustering problems is balanced clustering where we wish to obtain a partition with clusters of the same size. For example, DeSieno (1988) introduced a conscience mechanism which penalizes clusters relative to their size, Ahalt et al. (1990), then employed it to develop the Frequency Sensitive Competitive Learning (FSCL) algorithm. In Li et al. (2018), authors proposed to leverage the exclusive lasso on the $k$-means and `min-cut` problems to regulate the balance degree of the clustering results. In Chen et al. (2017), authors proposed a self-balanced `min-cut` algorithm for image clustering implicitly using exclusive lasso as a balance regularizer in order to produce balanced partitions. Lin et al. (2019) proposed a simplex algorithm to solve a minimum cost flow problem similar to $k$-means. Pei et al. (2020) proposes a clustering algorithm based on a unified framework of $k$-means and ratio-cut and balanced partitions. The time and space complexity of our method are both linear with respect to the number .Wu et al. (2021) explores a balanced graph-based clustering model, named exponential-cut, via redesigning the intercluster compactness based on an exponential transformation. Liu et al. (2022) proposes to introduce a novel balanced constraint to regularize the clustering results and constrain the size of clusters in spectral

---

[1] https://github.com/chakib401/OT-cut

clustering. Wang et al. (2023) introduces a discrete and balanced spectral clustering with scalability model that integrates the learning the continuous relaxation matrix and the discrete cluster indicator matrix into a single step. Nie et al. (2020) proposes to use balanced clustering algorithms to learn embeddings

## 2.2 Constrained Clustering

Some clustering approaches with generic size constraints, which can be seen as an extension of balanced clustering, also exist. In Zhu et al. (2010), a heuristic algorithm to transform size constrained clustering problems into integer linear programming problems was proposed. Authors in Ganganath et al. (2014) introduced a modified k-means algorithm which can be used to obtain clusters of preferred sizes. Clustering paradigms based on OT generally offer the possibility to set a target distribution for resulting partitions. In Nie et al. (2024), a parameter-insensitive min cut clustering with flexible size constraints is proposed. Genevay et al. (2019) proposed a deep clustering algorithm through optimal transport with entropic regularization. In Laclau et al. (2017); Titouan et al. (2020); Fettal et al. (2022), authors proposed to tackle co-clustering and biclustering problems using OT demonstrating good empirical performance.

## 2.3 Gomov-Wasserstein Graph Clustering

The Gromov-Wasserstein (GW) partitioning paradigm S-GWL (Xu et al., 2019) supposes that the Gromov-Wasserstein discrepancy can uncover the clustering structure of the observed source graph $\mathcal{G}$ when the target graph $\mathcal{G}_{dc}$ only contains weighted self-connected isolated nodes, this means that the adjacency matrix of $\mathcal{G}_{dc}$ is diagonal. The weights of this diagonal matrix as well as the source and target distribution are special functions of the node degrees. Their approach uses a regularized proximal gradient method as well as a recursive partitioning scheme and can be used in a multi-view clustering setting. The problem with this approach is its sensitivity to the hyperparameter setting which is problematic since it is an unsupervised method. Abrishami et al. (2020) proposes an OT metric with a component based view of partitioning by assigning cost proportional to transport distance over graph edges. Another approach, SpecGWL (Chowdhury & Needham, 2021) generalizes spectral clustering using Gromov-Wasserstein discrepancy and heat kernels but suffers from high computational complexity. Given a graph with $n$ node, its optimization procedure involves the computation of a gradient which is in $O(n^3 \log(n))$ and an eigendecompostion $O(n^3)$ and therefore is not usable for large scale graphs. Liu & Wang (2022) leverages the OT probability to seek the edges of the graph that characterizes the local nonlinear structure of the original feature. A recent approach (Yan et al., 2024) uses a spectral optimal transport barycenter model, which learns spectral embeddings by solving a barycenter problem equipped with an optimal transport discrepancy and guidance of data.

# 3 Preliminaries

In what follows, $\Delta^n = \{\mathbf{p} \in \mathbb{R}_+^n \,|\, \sum_{i=1}^n p_i = 1\}$ denotes the $n$-dimensional standard simplex. $\Pi(\mathbf{w}, \mathbf{v}) = \{\mathbf{Z} \in \mathbb{R}_+^{n \times k} | \mathbf{Z} \mathbb{1} = \mathbf{w}, \mathbf{Z}^\top \mathbb{1} = \mathbf{v}\}$ denotes the transportation polytope, where $\mathbf{w} \in \Delta^n$ and $\mathbf{v} \in \Delta^k$ are the marginals of the joint distribution $\mathbf{Z}$ and $\mathbb{1}$ is a vector of ones, its size can be inferred from the context. Matrices are denoted with uppercase boldface letters, and vectors with lowercase boldface letters. For a matrix $\mathbf{M}$, its $i$-th row is $\mathbf{m}_i$ and $m_{ij}$ is the $j$-th entry of row $i$. Tr refers to the trace of a square matrix. $||.||$ refers to the Frobenius norm.

## 3.1 Graph Cuts and Spectral Clustering

**Minimum-cut Problem.** Given an undirected graph $\mathcal{G} = (\mathcal{V}, \mathcal{E})$ with a weighted adjacency matrix $\mathbf{W} \in \mathbb{R}^{n \times n}$ with $n = |\mathcal{V}|$, a cut is a partition of its vertices $\mathcal{V}$ into two disjoint subsets $\mathcal{A}$ and $\bar{\mathcal{A}}$. The value of a cut is given by

$$\mathrm{cut}(\mathcal{A}) = \sum_{v_i \in \mathcal{A}, v_j \in \bar{\mathcal{A}}} w_{ij}. \tag{1}$$

The goal of the minimum $k$-cut problem is to find a partition $(\mathcal{A}_1, ..., \mathcal{A}_k)$ of the set of vertices $\mathcal{V}$ into $k$ different groups that is minimal in some metric. Intuitively, we wish for the edges between different subsets

to have small weights, and for the edges within a subset have large weights. Formally, it is defined as

$$\texttt{min-cut}(\mathbf{W}, k) = \min_{\mathcal{A}_1,\ldots,\mathcal{A}_k} \sum_{i=1}^{k} \operatorname{cut}(\mathcal{A}_i). \tag{2}$$

This problem can also be stated as a trace minimization problem by representing the resulting partition $\mathcal{A}_1, \ldots, \mathcal{A}_k$ using an assignment matrix $\mathbf{X}$ such that for each row $i$, we have that

$$x_{ij} = \begin{cases} 1 & \text{if vertex } i \text{ is in } \mathcal{A}_j, \\ 0 & \text{otherwise.} \end{cases} \tag{3}$$

This condition is equivalent to introducing two constraints which are $\mathbf{X} \in \{0,1\}^{n \times k}$ and $\mathbf{X}\mathbb{1} = \mathbb{1}$. The minimum $k$-cut problem can then be formulated as

$$\texttt{min-cut}(\mathbf{W}, k) = \min_{\substack{\mathbf{X} \in \{0,1\}^{n \times k} \\ \mathbf{X}\mathbb{1} = \mathbb{1}}} \operatorname{Tr}(\mathbf{X}^\top \mathbf{L} \mathbf{X}), \tag{4}$$

where $\mathbf{L} = \mathbf{D} - \mathbf{W}$ refers to the graph Laplacian of the graph $\mathcal{G}$ and $\mathbf{D}$ is the diagonal matrix of degree of $\mathbf{W}$, i.e., $d_{ii} = \sum_j w_{ij}$.

**Normalized $k$-Cut Problem.** In practice, solutions to the minimum $k$-cut problem do not yield satisfactory partitions due to the formation of small groups of vertices. Consequently, versions of the problem that take into account some notion of "size" for these groups have been proposed. The most commonly used one is normalized cut (Shi & Malik, 2000):

$$\texttt{ncut}(\mathbf{W}, k) = \min_{\mathcal{A}_1,\ldots,\mathcal{A}_k} \sum_{i=1}^{k} \frac{\operatorname{cut}(\mathcal{A}_i)}{\operatorname{vol}(\mathcal{A}_i)}, \tag{5}$$

where the volume can be conveniently written as $\operatorname{vol}(\mathcal{A}_i) = \mathbf{x}_i^T \mathbf{D} \mathbf{x}_i$. Another variant which is referred to as the ratio cut problem due to the different groups being normalized by their cardinality instead of their volumes:

$$\texttt{rcut}(\mathbf{W}, k) = \min_{\mathcal{A}_1,\ldots,\mathcal{A}_k} \sum_{i=1}^{k} \frac{\operatorname{cut}(\mathcal{A}_i)}{|\mathcal{A}_i|}, \tag{6}$$

where $|\mathcal{A}_i| = \mathbf{x}_i^T \mathbf{x}_i$. This variant can be recovered from the normalized graph cut problem by replacing $\mathbf{D}$ with $\mathbf{I}$ in the computation of the volume.

**Spectral Clustering.** A common approach to solving the normalized graph cut problems, spectral clustering, relaxes the partition constraints on $\mathbf{X}$ and instead considers a form of semi-orthogonality constraints. In the case of $\texttt{rcut}$, we have $\texttt{rcut}$ written as a trace optimization problem:

$$\texttt{rcut}(\mathbf{W}, k) = \min_{\substack{\mathbf{X} \in \mathbb{R}^{n \times k} \\ \mathbf{X}^\top \mathbf{X} = \mathbf{I}}} \operatorname{Tr}\left(\mathbf{X}^\top \mathbf{L} \mathbf{X}\right). \tag{7}$$

On the other hand for $\texttt{ncut}$, the partition matrix $\mathbf{X}$ is substituted with $\mathbf{H} = \mathbf{D}^{1/2}\mathbf{X}$ and a semi-orthogonality constraint is placed on this $\mathbf{H}$, i.e.,

$$\texttt{ncut}(\mathbf{W}, k) = \min_{\substack{\mathbf{H} \in \mathbb{R}^{n \times k} \\ \mathbf{H}^\top \mathbf{H} = \mathbf{I}}} \operatorname{Tr}\left(\mathbf{H}^\top \mathbf{D}^{-1/2} \mathbf{L} \mathbf{D}^{-1/2} \mathbf{H}\right). \tag{8}$$

A solution $\mathbf{H}$ for the $\texttt{ncut}$ problem is formed by stacking the first $k$-eigenvectors of the symmetrically normalized Laplacian $\mathbf{L}_s = \mathbf{D}^{-1/2}\mathbf{L}\mathbf{D}^{-1/2}$ as its columns, and then applying a clustering algorithm such as $k$-means on its rows and assign the original data points accordingly (Ng et al., 2001). The principle is the same for solving $\texttt{rcut}$ but instead using the unnormalized Laplacian.

### 3.2 Optimal Transport

**Discrete optimal transport.** The goal of the optimal transport problem is to find a minimal cost transport plan $\mathbf{X}$ between a source probability distribution of $\mathbf{w}$ and a target probability distribution $\mathbf{v}$. Here we are interested in the discrete Kantorovich formulation of OT (Kantorovich, 1942). When dealing with discrete probability distributions, said formulation is

$$\mathtt{OT}(\mathbf{M}, \mathbf{w}, \mathbf{v}) = \min_{\mathbf{X} \in \Pi(\mathbf{w}, \mathbf{v})} \langle \mathbf{M}, \mathbf{X} \rangle, \tag{9}$$

where $\langle ., . \rangle$ is the Frobenius product, $\mathbf{M} \in \mathbb{R}^{n \times k}$ is the cost matrix, and $m_{ij}$ quantifies the effort needed to transport a probability mass from $\mathbf{w}_i$ to $\mathbf{v}_j$. Regularization can be introduced to further speed up computation of OT. Examples include entropic regularization (Cuturi, 2013; Altschuler et al., 2017) and low-rank regularization (Scetbon & marco cuturi, 2022), as well as, other types of approximations (Quanrud, 2019; Jambulapati et al., 2019).

**Discrete Gromov-Wasserstein Discrepancy.** The discrete Gromov-Wasserstein (GW) discrepancy (Peyré et al., 2016) is an extension of optimal transport to the case where the source and target distributions are defined on different metric spaces:

$$\mathrm{GW}(\mathbf{M}, \bar{\mathbf{M}}, \mathbf{w}, \mathbf{v}) = \min_{\mathbf{X} \in \Pi(\mathbf{w}, \mathbf{v})} = \left\langle L(\mathbf{M}, \bar{\mathbf{M}}) \otimes \mathbf{X}, \mathbf{X} \right\rangle = \sum_{i,j,k,l} L(m_{ik}, \bar{m}_{jl}) x_{ij} x_{kl} \tag{10}$$

where $\mathbf{M} \in \mathbb{R}^{n \times n}$ and $\bar{\mathbf{M}} \in \mathbb{R}^{k \times k}$ are similarity matrices defined on the source space and target space respectively, and $L : \mathbb{R} \times \mathbb{R} \to \mathbb{R}$ is a divergence measure between scalars, $L(\mathbf{M}, \bar{\mathbf{M}})$ symbolizes the $n \times n \times k \times k$ tensor of all pairwise divergences between the elements of $\mathbf{M}$ and $\bar{\mathbf{M}}$. $\otimes$ denotes tensor-matrix product. Different approximation schemes have been explored for this problem Altschuler et al. (2018).

## 4 Proposed Methodology

In this section, we derive our OT-based constrained graph cut problem and propose a nonconvex proximal GD algorithm which guarantees global convergence to a critical point.

### 4.1 Normalized Cuts via Optimal Transport

As already mentioned, the good performance of the normalized cut algorithm comes from the normalization by the volume of each group in the cut. However, the size constraint is not a hard one, meaning that obtained groups are not of exactly the same volume. This leads us to propose to replace the volume normalization by a strict balancing constraint as follows:

$$\min_{\mathcal{A}_1, \dots, \mathcal{A}_k} \sum_{i=1}^{k} \mathrm{cut}(\mathcal{A}_i) \quad \text{s.t.} \quad \mathrm{vol}(\mathcal{A}_1) = \dots = \mathrm{vol}(\mathcal{A}_k). \tag{11}$$

this problem can be rewritten as the following trace minimization problem:

$$\min_{\mathbf{X}} \quad \mathrm{Tr}(\mathbf{X}^\top \mathbf{L} \mathbf{X})$$

subject to:

$$\begin{cases} \mathbf{X} \in \mathbb{R}_+^{n \times k} \\ \mathbf{X} \mathbb{1} = \mathbf{D} \mathbb{1}, & (\mathbf{x}_i \text{ sums to the degree of node } i) \\ \mathbf{X}^\top \mathbb{1} = \dfrac{\sum_i d_{ii}}{k} \mathbb{1}, & (\text{clusters are balanced w.r.t degrees}) \\ \forall_i \|\mathbf{x}_i\|_0 = 1 & (\text{a node belongs to a unique cluster.}) \end{cases} \tag{12}$$

Here, $\|.\|_0$ is the zero norm that returns the number of nonzero elements in its argument. This problem may not have feasible solutions. However, by dropping the fourth constraint, this problem becomes an instance

of the Gromov-Wasserstein problem with an $\ell_2$ loss which is always feasible. Specifically, the first, second and third constraints are equivalent to defining $\mathbf{X}$ to be an element of the transportation polytope with a uniform target distribution and a source distribution consisting of the degrees of the nodes. These degrees can be represented as proportions instead of absolute quantities by dividing them over their sum, yielding the following problem:

$$\min_{\mathbf{X}\in\Pi\left(\frac{1}{\sum_i d_{ii}}\mathbf{D}\mathbb{1},\frac{1}{k}\mathbb{1}\right)} \mathrm{Tr}(\mathbf{X}^\top\mathbf{L}\mathbf{X}) \tag{13}$$

This formulation is a special case of the Gromov-Wasserstein problem for a source space whose similarity matrix in the initial space is $\mathbf{M} = -\mathbf{L}$ and whose similarity matrix in the destination space is $\bar{\mathbf{M}} = \mathbf{I}$. Note that a ratio cut version can be obtained by replacing the volume constraint with

$$|\mathcal{A}_1| = \ldots = |\mathcal{A}_k| \tag{14}$$

in problem 11, and similarly in problem 13, by substituting the identity matrix $\mathbf{I}$ for the degree matrix $\mathbf{D}$, giving rise to:

$$\min_{\mathbf{X}\in\Pi\left(\frac{1}{n}\mathbb{1},\frac{1}{k}\mathbb{1}\right)} \mathrm{Tr}(\mathbf{X}^\top\mathbf{L}\mathbf{X}) \tag{15}$$

## 4.2 Graph Cuts with Arbitrary Size Constraints

From the previous problem, it is easy to see that target distribution does not need to be uniform, and as such, any distribution can be considered, leading to further applications like imbalanced dataset clustering. Another observation is that any notion of size can be considered and not only the volume or cardinality of the formed node groups. We formulate an initial version of the generic optimal transport graph cut problem as:

$$\min_{\mathbf{X}\in\Pi(\boldsymbol{\pi}^s,\boldsymbol{\pi}^t)} \mathrm{Tr}(\mathbf{X}^\top\mathbf{L}\mathbf{X}) \equiv \min_{\mathbf{X}\in\Pi(\boldsymbol{\pi}^s,\boldsymbol{\pi}^t)} \langle \mathbf{L}\mathbf{X}, \mathbf{X}\rangle, \tag{16}$$

where $\boldsymbol{\pi}_i^s$ is the relative 'size' of the element $i$ and $\boldsymbol{\pi}_j^t$ is the desired relative 'size' of the group $j$. Through the form that uses the Frobenius product, it is easy to see how our problem is related to the Gromov-Wasserstein problem.

## 4.3 Regularization for Sparse Solutions

We wish to obtain sparse solutions in order to easily interpret them as partition matrices of the input graph. We do so by aiming to find solutions over the extreme points of the transportation polytope which are matrices that have at most $n + k - 1$ non-zero entries (Peyré et al., 2019). We do so by introducing a regularization term to problem 16. Consequently, we consider the following problem which we coin `OT-cut`:

$$\texttt{OT-cut}(\mathbf{X},\pi_s,\pi_t) \equiv \min_{\mathbf{X}\in\Pi(\boldsymbol{\pi}^s,\boldsymbol{\pi}^t)} \mathrm{Tr}(\mathbf{X}^\top\mathbf{L}\mathbf{X}) - \lambda\|\mathbf{X}\|^2 \tag{17}$$

where $\lambda \in \mathbb{R}^+$ is the regularization trade-off parameter. It should be noted that our regularizer is concave. We also define two special cases of this problem, which are based on the `ncut` and `rcut` problems. The first one which we call `OT-ncut` is obtained by fixing the hyper-parameter $\pi_s = \frac{1}{\sum_i d_{ii}}\mathbf{D}\mathbb{1}$ while the second one `OT-rcut` is obtained by substituting the $\mathbf{D}$ in the previous formula with $\mathbf{I}$ and forcing the target to be uniform. Figure 1 shows the evolution of the objective on different datasets.

## 4.4 Optimization, Convergence and Complexity

We wish to solve problem 17 which is nonconvex, but algorithms with convergence guarantees exist for problems of this form. Specifically, we will be using a nonconvex proximal gradient descent based on Li & Lin (2015). The pseudocode is given in algorithm 1.

**Proposition 1.** *For step size $\alpha = \frac{1}{2\lambda}$, the iterates $\mathbf{X}^{(t)}$ generated by the nonconvex PGD algorithm for our problem are all extreme points of the transportation polytope, and as such, have at most $n + k - 1$ nonzero entries.*

*Proof.* Problem 17 can be equivalently stated by writing the constraint as a term in the loss function:

$$\min_{\mathbf{X}} \quad \underbrace{\mathrm{Tr}(\mathbf{X}^\top \mathbf{L} \mathbf{X})}_{f(\mathbf{X})} + \underbrace{I_{\Pi(\boldsymbol{\pi}^s, \boldsymbol{\pi}^t)}(\mathbf{X}) - \lambda \|\mathbf{X}\|^2}_{g(\mathbf{X})} \tag{18}$$

where $I_{\mathcal{C}}$ is the characteristic function of set $\mathcal{C}$ i.e.

$$I_{\mathcal{C}} = \begin{cases} 0, & \text{if } \mathbf{X} \in \mathcal{C}, \\ +\infty, & \text{if } \mathbf{X} \notin \mathcal{C}. \end{cases}$$

Since we use a proximal descent scheme, we show how to compute the proximal operator for our loss function:

$$\mathrm{prox}_{\alpha g}\left(\mathbf{X}^{(t)} - \alpha \nabla f(\mathbf{X}^{(t)})\right) = \mathrm{prox}_{\alpha g}\left(\mathbf{X}^{(t)} - \alpha \nabla \mathrm{Tr}\left(\mathbf{X}^{(t)} \mathbf{L} \mathbf{X}^{(t)}\right)\right)$$

$$= \mathrm{prox}_{\alpha(I_{\Pi(\boldsymbol{\pi}^s, \boldsymbol{\pi}^t)} - \alpha\|.\|^2)}\left((\mathbf{I} - 2\alpha\mathbf{L})\mathbf{X}^{(t)}\right) = \underset{\mathbf{Z} \in \Pi(\boldsymbol{\pi}^s, \boldsymbol{\pi}^t)}{\arg\min} \frac{1}{2\alpha}\left\|\mathbf{Z} - (\mathbf{I} - 2\alpha\mathbf{L})\mathbf{X}^{(t)}\right\|^2 - \lambda\|\mathbf{Z}\|^2$$

$$= \underset{\mathbf{Z} \in \Pi(\boldsymbol{\pi}^s, \boldsymbol{\pi}^t)}{\arg\min} \frac{1}{2\alpha}\|\mathbf{Z}\|^2 + \frac{1}{2\alpha}\left\|(\mathbf{I} - 2\alpha\mathbf{L})\mathbf{X}^{(t)}\right\|^2 - \frac{1}{\alpha}\mathrm{Tr}\left(\mathbf{Z}^\top(\mathbf{I} - 2\alpha\mathbf{L})\mathbf{X}^{(t)}\right) - \lambda\|\mathbf{Z}\|^2.$$

We assumed that $\alpha = \frac{1}{2\lambda}$, by substituting for $\lambda$ into the previous formula and dropping the constant term, we obtain:

$$\underset{\mathbf{Z} \in \Pi(\boldsymbol{\pi}^s, \boldsymbol{\pi}^t)}{\arg\min} \mathrm{Tr}\left(\mathbf{Z}^\top(2\alpha\mathbf{L} - \mathbf{I})\mathbf{X}^{(t)}\right) = \underset{\mathbf{Z} \in \Pi(\boldsymbol{\pi}^s, \boldsymbol{\pi}^t)}{\arg\min} \left\langle\mathbf{Z}, \ (2\alpha\mathbf{L} - \mathbf{I})\mathbf{X}^{(t)}\right\rangle.$$

This is the classical OT problem. Its resolution is possible by stating it as the earth-mover's distance (EMD) linear program (Hitchcock, 1941) and using the network simplex algorithm (Bonneel et al., 2011). □

**Proposition 2.** *Algorithm 1 globally converges for step size $\alpha < \frac{1}{s}$ where $s$ is the smoothness constant of $Tr(\mathbf{X}^\top \mathbf{L} \mathbf{X})$.*

*Proof.* Here, we have that $f$ is proper and $s$-smooth i.e. $\nabla f$ is $s$-Lipschitz. $g$ is proper and lower semi-continuous. Additionally, $f + g$ is coercive. Then, according to theorem 1 in Li & Lin (2015), nonconvex accelerated proximal GD globally converges for $\alpha < \frac{1}{s}$. □

**Proposition 3.** *For a graph with $n$ nodes, the complexity of an iteration of the proposed algorithm is $\mathcal{O}\left(kn^2 \log n\right)$.*

*Proof.* We note that in practice $n >> k$ and that the complexity of the network simplex algorithm for some graph $\mathcal{G}_{EMD} = (\mathcal{V}_{EMD}, \mathcal{V}_{EMD})$ is in $\mathcal{O}(|\mathcal{V}_{EMD}||\mathcal{E}_{EMD}| \log |\mathcal{E}_{EMD}|)$ (Orlin, 1997). In our case, this graph has $|\mathcal{V}_{EMD}| = n + k$ (since $n >> k$, we can drop the $k$) and $|\mathcal{E}_{EMD}| = nk$. The other operation that is performed during each iteration is the matrix multiplication whose complexity is in $\mathcal{O}(k|\mathcal{E}|)$ where $|\mathcal{E}|$ is the number of edges in the original graph. In the worst case when matrix $\mathbf{L}$ is fully dense, we have that $|\mathcal{E}| = n^2$. □

# 5 Experiments

We evaluated the clustering performance of our two variants `OT-ncut` and `OT-rcut` algorithms against the spectral clustering algorithm and state-of-the-art OT-based graph clustering approaches.

## 5.1 Datasets

We perform experiments on graphs constructed from image datasets, namely, MNIST (Deng, 2012), Fashion-MNIST (Xiao et al., 2017) and KMNIST (Clanuwat et al., 2018). We generate these graphs using three subspace clustering approaches: low-rank subspace clustering (LRSC) (Vidal & Favaro, 2014), least-square regression subspace clustering (LSR) (Lu et al., 2012) and elastic net subspace clustering (ENSC) (You

---

**Algorithm 1:** Nonconvex Accelerated PGD for `OT-cut`

---

**Data:** $\mathbf{A}$ Adjacency matrix, $\boldsymbol{\pi}^s$ node size distribution, $\boldsymbol{\pi}^t$ cluster size distribution, $\mathbf{G}_{init}$ initial partition matrix, $\alpha = \frac{1}{2\lambda} < \frac{1}{s}$ step size, *maxIter* maximum number of iterations.

**Result:** $\mathbf{G}$ partition of the graph.

Construct Laplacian matrix $\mathbf{L}$ from the adjacency matrix $\mathbf{A}$;

$\mathbf{X}^{(0)} \leftarrow \arg \mathtt{OT}\left(\mathbf{G}_{init}, \boldsymbol{\pi}^s, \boldsymbol{\pi}^t\right)$;

$\mathbf{Z}^{(1)} \leftarrow \mathbf{X}^{(0)}, \mathbf{X}^{(1)} \leftarrow \mathbf{X}^{(0)}$;

$c_0 \leftarrow 0, c_1 \leftarrow 1$;

**while** *maxIter not reached* **do**

$\quad \mathbf{Y}^{(t)} = \mathbf{X}^{(t)} + \frac{c_{t-1}}{c_t}(\mathbf{Z}^{(t)} - \mathbf{X}^{(t)}) + \frac{c_{t-1}-1}{c_t}(\mathbf{X}^{(t)} - \mathbf{X}^{(t-1)})$;

$\quad \mathbf{Z}^{(t+1)} := \arg \mathtt{OT}\left((2\alpha\mathbf{L} - \mathbf{I})\mathbf{Y}^{(t)}, \boldsymbol{\pi}^s, \boldsymbol{\pi}^t\right)$;

$\quad \mathbf{V}^{(t+1)} := \arg \mathtt{OT}\left((2\alpha\mathbf{L} - \mathbf{I})\mathbf{X}^{(t)}, \boldsymbol{\pi}^s, \boldsymbol{\pi}^t\right)$;

$\quad c_{t+1} = (\sqrt{4c_t^2 + 1} + 1)/2$;

$\quad \mathbf{X}^{(t+1)} = \begin{cases} \mathbf{Z}^{(t+1)}, & \text{if loss}\left(\mathbf{Z}^{(t+1)}\right) < \text{loss}\left(\mathbf{V}^{(t+1)}\right) \\ \mathbf{V}^{(t+1)}, & \text{otherwise.} \end{cases}$ ;

**end**

Generate partition matrix $\mathbf{G}$ by assigning each node $i$ to the $(\arg \max_i \mathbf{x}_i)$-th partition.;

---

Table 1: Dataset Statistics. The balance ratio is the ratio of the most frequent class over the least frequent one.

| Type | Dataset | Nodes | Graph & Edges | Sparsity | Clusters | Balance Ratio |
|---|---|---|---|---|---|---|
| | | | LRSC (100,000,000) | 0.0% | | |
| | MNIST | 10,000 | LSR (100,000,000) | 0.0% | 10 | 1.272 |
| | | | ENSC (785,744) | 99.2% | | |
| Graphs Built From Images | | | LRSC (100,000,000) | 0.0% | | |
| | Fashion-MNIST | 10,000 | LSR (100,000,000) | 0.0% | 10 | 1.0 |
| | | | ENSC (458,390) | 99.5% | | |
| | | | LSR (100,000,000) | 0.0% | | |
| | KMNIST | 10,000 | LRSC (100,000,000) | 0.0% | 10 | 1.0 |
| | | | ENSC (817,124) | 99.2% | | |
| | ACM | 3,025 | 2,210,761 | 75.8% | 3 | 1.099 |
| Naturally Occuring Graphs | DBLP | 4,057 | 6,772,278 | 58.9% | 4 | 1.607 |
| | Village | 1,991 | 16,800 | 99.6% | 12 | 3.792 |
| | EU-Email | 1,005 | 32,770 | 96.8% | 42 | 109.0 |

et al., 2016). We also consider four graph datasets: DBLP, a co-term citation network; and ACM, a co-author citation networks (Fan et al., 2020). EU-Email an email network from a large European research institution (Leskovec & Krevl, 2014). Indian-Village describes interactions among villagers in Indian villages (Banerjee et al., 2013). The statistical summaries of these datasets are available in Table 1.

Table 2: Average (±sd) clustering performance and running times on the graph built from images. The best performance is highlighted in bold, the lowest (highest) runtime is highlighted in blue (red).

| Graph | Method | MNIST | | Fashion-MNIST | | KMNIST | |
|---|---|---|---|---|---|---|---|
| | | ARI | Time | ARI | Time | ARI | Time |
| LRSC | Spectral | 0.4134 ±0.0003 | 10.28 | 0.1742 ±0.0003 | 10.51 | 0.4067 ±0.0 | 9.83 |
| | S-GWL | 0.0488 ±0.0 | 7.88 | 0.0188 ±0.0 | 7.84 | 0.0560 ±0.0 | 7.98 |
| | SpecGWL | 0.0248 ±0.0 | 453.19 | 0.0111 ±0.0 | 397.19 | 0.0145 ±0.0 | 383.23 |
| | **OT-rcut** | 0.4516 ±0.0273 | 5.58 | 0.2231 ±0.0051 | 5.82 | **0.4157** ±0.0154 | 6.15 |
| | **OT-ncut** | **0.4751** ±0.0383 | 6.12 | **0.2291** ±0.0148 | 6.11 | 0.3832 ±0.0279 | 5.88 |
| LSR | Spectral | 0.311 ±0.0002 | 8.82 | 0.1486 ±0.0001 | 10.19 | 0.3631 ±0.0001 | 9.72 |
| | S-GWL | 0.0628 ±0.0 | 8.2 | 0.0357 ±0.0 | 7.93 | 0.0593 ±0.0 | 8.01 |
| | SpecGWL | 0.1127 ±0.0 | 454.06 | 0.0341 ±0.0 | 454.26 | 0.0267 ±0.0 | 407.55 |
| | **OT-rcut** | **0.3723** ±0.0377 | 6.23 | 0.1771 ±0.0164 | 6.61 | **0.4335** ±0.0105 | 6.01 |
| | **OT-ncut** | 0.3458 ±0.0267 | 5.77 | **0.1915** ±0.0131 | 5.72 | 0.4301 ±0.0075 | 5.87 |
| ENSC | Spectral | 0.1206 ±0.0001 | 13.28 | 0.1164 ±0.0 | 12.62 | **0.4321** ±0.0007 | 13.6 |
| | S-GWL | 0.0798 ±0.0 | 8.05 | 0.0362 ±0.0 | 7.85 | 0.0422 ±0.0 | 7.96 |
| | SpecGWL | **0.5444** ±0.0 | 268.12 | 0.1082 ±0.0 | 288.75 | 0.4020 ±0.0 | 287.06 |
| | **OT-rcut** | 0.4228 ±0.0694 | 5.47 | 0.2113 ±0.0257 | 6.18 | 0.2924 ±0.0589 | 6.27 |
| | **OT-ncut** | 0.3882 ±0.0718 | 5.68 | **0.2251** ±0.0191 | 5.77 | 0.2771 ±0.0226 | 5.83 |

Table 3: Average (±sd) clustering performance and running times on the graph datasets. Same legend as for Table 2.

| Method | EU-Email | | Village | | ACM | | DBLP | |
|---|---|---|---|---|---|---|---|---|
| | ARI | Time | ARI | Time | ARI | Time | ARI | Time |
| Spectral | 0.2445 ±0.0133 | 1.13 | 0.3892 ±0.1934 | 0.76 | 0.1599 ±0.003 | 1.83 | 0.0039 ±0.0053 | 16.49 |
| Greedy | 0.1711 ±0.0 | 2.28 | 0.0002 ±0.0 | 1.15 | 0.0 ±0.0 | 37.59 | 0.1375 ±0.0 | 144.21 |
| Infomap | **0.3087** ±0.0 | 0.09 | — | | — | | -0.0001 ±0.0 | 30.71 |
| S-GWL | 0.2684 ±0.0 | 2.11 | 0.5333 ±0.0 | 4.26 | 0.1873 ±0.0 | 2.09 | 0.0 ±0.0 | 18.42 |
| SpecGWL | 0.1125 ±0.0 | 0.59 | 0.5887 ±0.0 | 0.89 | 0.008 ±0.0 | 4.65 | 0.2891 ±0.0 | 13.66 |
| **OT-rcut** | 0.2629 ±0.0096 | 0.22 | **0.5969** ±0.0505 | 0.27 | **0.2643** ±0.0249 | 0.69 | **0.3119** ±0.0279 | 1.45 |
| **OT-ncut** | 0.2687 ±0.0094 | 0.20 | 0.4819 ±0.0369 | 0.30 | 0.2167 ±0.045 | 0.77 | 0.1721 ±0.0674 | 1.33 |

## 5.2 Performance Metrics

We adopt Adjusted Rand Index (ARI) (Hubert & Arabie, 1985) to evaluate clustering performance. It takes values between 1 and -0.5; larger values signify better performance. To evaluate the concordance of the desired and the obtained cluster distributions, we use the Kullback-Leibler (KL) divergence (Kullback & Leibler, 1951). The KL divergence between two perfectly matching distributions will be equal to zero. Otherwise, it would be greater than zero. Smaller KL values signify better concordance.

## 5.3 Experimental settings

Our two variants, `OT-ncut` and `OT-rcut` are implemented via the Python optimal transport package (POT) (Flamary et al., 2021). We use random initialization and use uniform target distributions unless explicitly stated otherwise. We also set $\alpha = 1/2$ and the number of iterations to 30 for the image graphs and 20 for the other graphs. We also use normalized laplacian matrices. For the baselines, we use the Scikit-Learn (Pedregosa et al., 2011) implementation of spectral clustering. We use the official implementations of S-GWL (Xu et al., 2019) and SpecGWL (Chowdhury & Needham, 2021). Furthermore, we considered the baselines

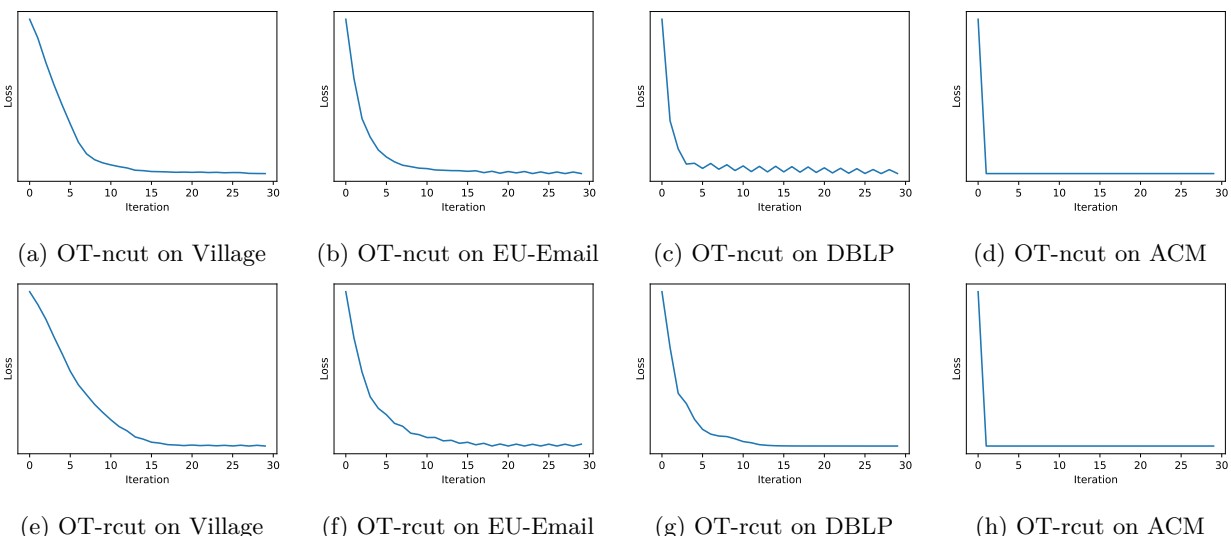

Figure 1: Evolution of the objective as function of the number of iterations.

Table 4: Faithfulness to the constraints: KL divergence between the desired and the resulting cluster distributions. A value of zero reflects a perfect match between the constraint and the result.

| Dataset | Graph | OT-rcut | OT-ncut |
|---|---|---|---|
| MNIST | LRSC | 0.0 ±0.0 | 0.0001 ±0.0001 |
| | LSR | 0.0 ±0.0 | 0.0001 ±0.0001 |
| | ENSC | 0.0 ±0.0 | 0.0 ±0.0 |
| Fashion-MNIST | LRSC | 0.0 ±0.0 | 0.0001 ±0.0001 |
| | LSR | 0.0 ±0.0 | 0.0001 ±0.0001 |
| | ENSC | 0.0 ±0.0 | 0.0 ±0.0 |

| Dataset | Graph | OT-rcut | OT-ncut |
|---|---|---|---|
| KMNIST | LRSC | 0.0 ±0.0 | 0.0 ±0.0 |
| | LSR | 0.0 ±0.0 | 0.0001 ±0.0001 |
| | ENSC | 0.0 ±0.0 | 0.0 ±0.0 |
| ACM | | 0.0 ±0.0 | 0.0 ±0.0 |
| DBLP | | 0.0 ±0.0 | 0.00.0 ±0.0021 |
| Village | | 0.0 ±0.0 | 0.0011 ±0.0027 |
| EU-Email | | 0.0 ±0.0 | 0.0004 ±0.0007 |

used in Chowdhury & Needham (2021), namely, Fluid (Parés et al., 2018), Louvain (Blondel et al., 2008), Infomap (Rosvall et al., 2009) and Greedy (Clauset et al., 2004):

1. We reported results on the naturally occuring graphs only due to excessive run times over the image graphs.

2. Louvain and Infomap do not allow to specify the number of clusters. Comparison between partitions with a different number of clusters using ARI is not meaningful. As such, we only reported results on datasets for which those algorithms manage to recover the ground truth-number of clusters (for all runs). Louvain was dropped since it never managed to find the ground-truth number of clusters.

3. Fluid was dropped because it requires graphs that have a single connected component. This is not the case for any of the naturally occurring graphs.

4. We use the implementations of Louvain, Fluid, and Greedy provided in the networkx package (Hagberg et al., 2008). We use the implementation of Infomap provided in Edler et al. (2024).

All experiments were run five times and were performed on a 64gb RAM machine with a 12th Gen Intel(R) Core(TM) i9-12950HX (24 CPUs) processor with a frequency of 2.3GHz.

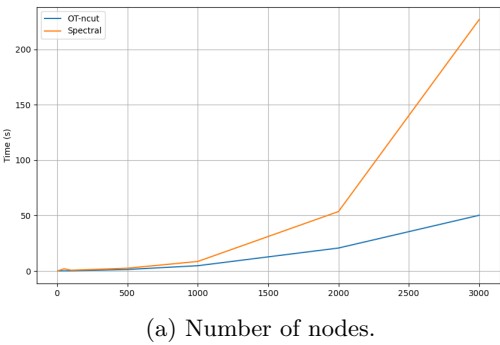 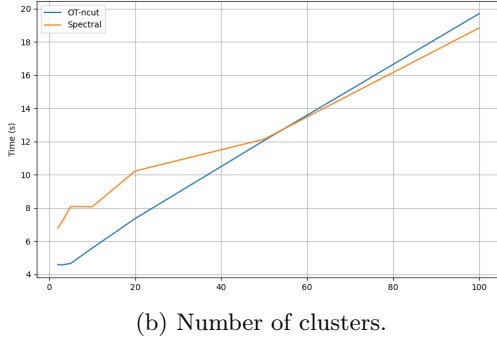

(a) Number of nodes.                    (b) Number of clusters.

Figure 2: Running times of Spectral clustering and OT-ncut on subsets of MNIST as a function of the number of nodes and the number of clusters.

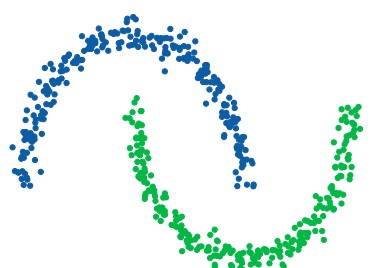 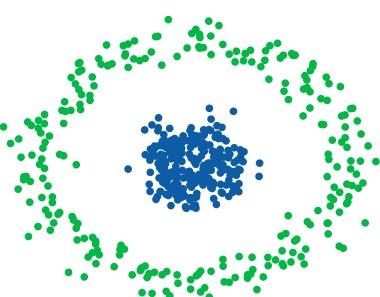

(a) OT-ncut on a a graph built using a k-nn graph.    (b) OT-rcut on a graph built using the RBF kernel.

Figure 4: OT-ncut and OT-rcut results on toy datasets.

### 5.4 Results

**Toy Datasets.** Our algorithm deals with a graph cut-like criterion which means that it should partition a dataset according to its connectivity. This means that it should work on datasets on which metric clustering approaches such as k-means fail. Two toy examples are given in Figure 3a and Figure 3b.

**Clustering Performance.** Table 3 presents the clustering performance on the graph datasets. In all cases, one of our two variants has the best results in terms of ARI except on EU-Email where Infomap has the best performance. Table 2 describes results obtained on image graph datasets. One of our two variants gives the best results on all three datasets with the graphs generated by LRSC and LSR. On the graphs generated by ENSC, the best result is obtained only on Fashion-MNIST while SpecGWL has the best results on MNIST. Spectral clustering gives the best performance on KMNIST. Note that better results can also be obtained with our variants by trading-off some computational efficiency. Specifically, this can be done by using several different initializations and taking the one that leads to minimizing the objective the most.

**Imbalanced Datasets.** Results on long-tailed versions of CIFAR-10 are reported in table 5. We notice that using ground truth cluster distribution constraints leads to better results when comparing to the traditional spectral clustering algorithm.

**Statistical Significance Testing** Figure 5 shows the performance ranks of the different methods averaged over all the runs on the datasets we considered in terms of ARI. The Neményi post-hoc rank test (Nemenyi, 1963) shows that OT-rcut and OT-ncut perform similarly and outperform the other approaches for a confidence level of 95%. Other approaches perform similarly.

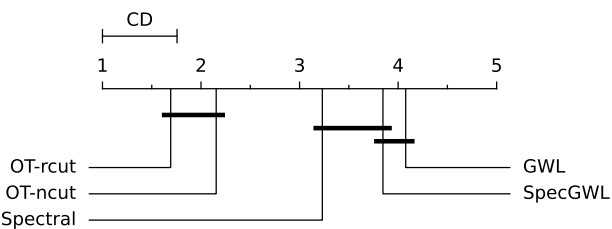

Figure 5: Neményi post-hoc rank test results. OT-rcut and OT-ncut outperform the baselines in a statistically significant manner.

Table 5: Image clustering performance in terms of ARI on the long-tailed CIFAR-10 datasets. Values are the averages over five runs.

| Balance | 5 | 10 | 50 | 100 |
|---|---|---|---|---|
| Spectral | 0.0566 | 0.0622 | 0.0584 | 0.0682 |
| OT-ncut | **0.0831** | 0.0730 | **0.0794** | **0.0693** |
| OT-rcut | 0.0731 | **0.0779** | 0.0752 | 0.0574 |

**Concordance of the Desired & Resulting Cluster Sizes.** To evaluate our algorithm's ability to produce a partition with the desired group size distribution, we use the KL divergence metric. Specifically, we compare the distribution obtained by our OT-rcut and OT-ncut variants against the target distribution specified as a hyperparameter ($\boldsymbol{\pi}^t$). Table 4 presents the KL divergences for both variants on various datasets. Predictably, our approaches achieve near-perfect performance on most datasets. Notably, OT-rcut is always able to perfectly recover the desired group sizes. This has to do with the fact that, up to a constant, all the entries in the solutions to the `rcut` problem are integers. This is not necessarily the case for `ncut` but the KL divergence is still very small due to the sparsity of the solutions.

**Running Times.** As shown in Table 3 and Table 2, OT-ncut and OT-rcut are the fastest in terms of execution times compared to other approaches on all datasets. As the graphs got larger, SpecGWL consistently had the largest runtimes. We also report the running times of spectral clustering and our OT-ncut approach on subsets of increasing size of MNIST as well as for increasing numbers of clusters in fig 2. The efficiency of our approach becomes increasingly significant compared to spectral clustering as the number of nodes grows. However, spectral clustering matches our approach's efficiency as the number of clusters increases. Our approach can be made more efficient by adopting sparse representations of the optimal transport plans when doing matrix multiplication.

# 6 Conclusion

In this paper we proposed a new graph cut algorithm for partitioning with arbitrary size constraints through optimal transport. This approach generalizes the concept of the normalized and ratio cut to arbitrary size distributions to any notion of size. We derived an algorithm that results in sparse solutions and guarantees global convergence to a critical point. Experiments on balanced and imbalanced datasets showed the superiority of our approach both in terms of clustering performance and empirical execution times compared to spectral clustering and other OT-based graph clustering approaches. They also demonstrated our approach's ability to recover partitions that match the desired ones which is valuable for practical problems where we wish to obtain balanced or constrained partitions.

# 7 Limitations

The node and cluster size distribution parameters can either be set using prior domain knowledge or through tuning them by trying different possible values and then selecting the best one via internal clustering quality metrics such as Davies-Bouldin index (Davies & Bouldin, 1979). In cases where no domain knowledge exits and parameter tuning is impossible, we can weigh each node by its degree and give clusters uniform sizes. This option is similar to what is done normalized cuts where no prior knowledge on size distributions is explicitly available Shi & Malik (2000). This issue will be studied in future works.

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

# A List of symbols

A non-exhaustive list of the symbols used throughout the paper is available in table 6.

Table 6: List of symbols.

| Symbol | Description |
|---|---|
| $\mathcal{G} = (\mathcal{V}, \mathcal{E})$ | A graph with a set of vertices $\mathcal{V}$ and edges $\mathcal{E}$. |
| $\mathcal{A}_1, \ldots, \mathcal{A}_k$ | A partition of the nodes of graph $\mathcal{G}$ |
| $\bar{\mathcal{A}}$ | The complementary set of $\mathcal{A}$ |
| $\mathbf{A}$ | Adjacency matrix of $\mathbf{G}$ |
| $\mathbf{D}$ | Diagonal matrix of degrees of $\mathcal{G}$ |
| $\mathbf{L}$ | Laplacian matrix of $\mathcal{G}$ |
| $\mathbf{X}$ | A partition matrix or a transport plan depending on context |
| $\text{vol}(.)$ | Volume of a set of nodes |
| $|.|$ | Cardinality of a set |
| $\text{Tr}$ | Trace operator |
| $< ., . >$ | Frobenius product |
| $\|.\|$ | Frobenius norm |
| $\|.\|_0$ | Zero norm |
| $\otimes$ | Tensor-matrix product |
| $\Pi$ | A transportation polytope |
| $\mathbf{M}, \bar{\mathbf{M}}$ | Similarity matrices |
| $L(\mathbf{M}, \bar{\mathbf{M}})$ | The tensor of all pairwise divergences between the elements of $\mathbf{M}$ and $\bar{\mathbf{M}}$ |
| $\mathbf{G}$ | A partition matrix |
| $\mathbf{Y}, \mathbf{Z}, \mathbf{Y}$ | Transport plans |
| $\pi^s, \pi^t$ | Probability distributions |
| $I_{\mathcal{C}}$ | The characteristic function of $\mathcal{C}$ |
| $\mathbb{1}$ | Vector of ones |
| $c_t, s, \alpha, \lambda$ | Scalars |

