# OpenReview forum: "Graph Cuts with Arbitrary Size Constraints Through Optimal Transport"
_TMLR — Accepted by TMLR_

### Review · Reviewer_aAB4 · 2024-05-30

**Summary Of Contributions:**

In this paper, the authors introduce a novel perspective on graph partitioning, utilizing optimal transport theory within a graph cut framework with arbitrary size constraints. They frame the classic graph cut problem as a Gromov-Wasserstein problem and apply a proximal gradient descent algorithm for its resolution. Experimental results on datasets with diverse balance levels demonstrate the effectiveness of the proposed approach.

**Audience:**

Yes

**Broader Impact Concerns:**

The broader impact section is not included in this article. Based on my assessment, there are no concerns regarding the ethical implications of this work that need to be addressed.

**Claims And Evidence:**

No

**Requested Changes:**

Please address the weaknesses mentioned above when revising the article. Please ensure that the revised version effectively resolves all the mentioned issues.

**Strengths And Weaknesses:**

Strengths:
1. Considering the flexibility of cluster sizes in practical applications, the optimal transport theory is applied to spectral clustering, leading to a spectral clustering framework with arbitrary cluster size constraints.

2. In experiments, statistical significance testing is conducted to verify that the proposed approach significantly outperforms comparison methods. The KL divergence between the resulting and desired cluster size distributions is examined to confirm the proposed approach's ability to generate promising cluster size distributions.

Weaknesses:
1. The proposed model is not entirely accurate. A constraint \mathbf{X} > \mathcal{0} should be imposed on the transportation matrix \mathbf{X}. Without this constraint, the conclusion that \nabla f is s-Lipschitz cannot be guaranteed, and the partition matrix \mathbf{G} cannot be obtained by assigning each node i to the (\arg\max_i \mathbf{x}_i)-th partition.

2. The selection of parameters in the proposed model significantly influences its effectiveness. While size parameters can be set using domain knowledge, this requires strong prior knowledge of the dataset's size distribution, which is impractical in real-world unsupervised applications. If these size parameters are determined using other algorithms, the performance of the proposed model will be heavily influenced by those algorithms. Thus, the question arises: how can size parameters be determined in a reasonable manner, and what method was used in this paper? Additionally, the regularization parameter \lambda significantly affects the sparsity of the transportation matrix \mathbf{X}. Specifically, a smaller \lambda results in a sparser \mathbf{X}. Why is \lambda set to 1 when \alpha is set to 1/2? If \lambda is set to 1, can it be guaranteed that the nonzero entries of the matrix \mathbf{X} be n + k - 1?

3. The theoretical proofs are neither complete nor entirely correct. In the proof of Proposition 1, the constrained problem (17) is transformed into the regularized problem (18). A key issue is whether this transformation is indeed equivalent. If the constraints in problem (17) are moved to the objective as a regularization, a regularization parameter must be added, which will determine the equivalence relationship between problems (17) and (18). The proof of Proposition 1 seems to suggest that, since problem (17) can be transformed into the form of classical optimal transport, the n + k - 1 nonzero entries of matrix \mathbf{X} can be guaranteed. This is not entirely accurate. The sparsity of matrix \mathbf{X} is also influenced by the selection of the parameter \lambda. Moreover, in Proposition 2, it should be stated that f + g is convex rather than concave, and the proof is notably incomplete. It is not straightforward to conclude that the nonconvex proximal GD algorithm can globally converge in a special case based solely on the literature by Li & Lin (2015). Detailed supplementary proof is necessary.

4. The optimization algorithm is incomplete. The optimization procedure provided in the proof of Proposition 1 lacks sufficient detail for readers to fully understand it. Furthermore, the optimization procedure does not match the steps listed in Algorithm 1. Additionally, there is no explanation on how to initialize the partition matrix \mathbf{G}.

5. The literature review is incomplete. For balanced clustering, constrained clustering, and Gromov-Wasserstein graph clustering, numerous related and recent approaches exist. For instance, the following literature should be reviewed for balanced clustering:
[1] Nie, F., Zhu, W. and Li, X., 2020. Unsupervised large graph embedding based on balanced and hierarchical k-means. IEEE Transactions on Knowledge and Data Engineering, 34(4), pp.2008-2019.
[2] Pei, S., Nie, F., Wang, R. and Li, X., 2020. Efficient clustering based on a unified view of k-means and ratio-cut. Advances in Neural Information Processing Systems, 33, pp.14855-14866.
[3] Wu, D., Nie, F., Lu, J., Wang, R. and  Li, X., 2021. Balanced graph cut with exponential inter-cluster compactness. IEEE Transactions on Artificial Intelligence, 3(4), pp.498-505.
[4] Liu, C., Nie, F., Wang, R. and Li, X., 2022. Graph based soft-balanced fuzzy clustering. IEEE Transactions on Fuzzy Systems.
[5] Wang, R., Chen, H., Lu, Y., Zhang, Q., Nie, F. and  Li, X., 2023. Discrete and Balanced Spectral Clustering With Scalability. IEEE Transactions on Pattern Analysis and Machine Intelligence.
Additionally, there are some studies related to this work:
[6] Nie, F., Xie, F., Yu, W. and  Li, X., 2024. Parameter-insensitive Min Cut Clustering with Flexible Size Constrains. IEEE Transactions on Pattern Analysis and Machine Intelligence.
[7] Liu, S. and Wang, H., 2022. Graph convolutional optimal transport for hyperspectral image spectral clustering. IEEE Transactions on Geoscience and Remote Sensing, 60, pp.1-13.
[8] Yan, Y., Xu, Z., Yang, C., Zhang, J., Cai, R. and Ng, M. K. P., 2024, March. An Optimal Transport View for Subspace Clustering and Spectral Clustering. In Proceedings of the AAAI Conference on Artificial Intelligence (Vol. 38, No. 15, pp. 16281-16289).
[9] Chowdhury, S. and Needham, T., 2021, March. Generalized spectral clustering via Gromov-Wasserstein learning. In International Conference on Artificial Intelligence and Statistics (pp. 712-720). PMLR.

6. The experiments are neither complete nor convincing enough. The experimental settings are not described in sufficient detail to reproduce the reported results. It is recommended that the proposed approach be compared with more related works. The clustering performance is evaluated using only a single metric, ARI, and its value range is [-1, 1] rather than [-0.5, 1]. Additionally, the convergence results shown in Figure 1 clearly indicate that the proposed algorithm fails to converge. Why can't the convergence of the proposed algorithm be guaranteed?

7. There are many unclear descriptions in this paper. All abbreviations should be accompanied by their full terms upon first use. Additionally, why is the proximal GD algorithm referred to as the accelerated proximal GD algorithm? What aspect reflects the acceleration? There are many variables that are not mentioned in the notation section. In formula (9), what does “<>” mean? Several relevant references are not cited, such as those for discrete optimal transport, the Kantorovich formulation, and the network simplex algorithm. The statements "the proposed algorithm has globally convergence guarantees" in the abstract and conclusion are incorrect, as this is only valid in a special case, and the proof is incomplete.

8. There are numerous typos in this paper. In formula (6), "W" should be in bold. Below formula (9), "c_{ij}" should be "m_{ij}". In formula (12), below the notation "\min", the optimization variable \mathbf{X} should be indicated. In the proof of Proposition 2, "coercive" should be "concave".

9. The overall language expression in this article requires further revision. The current version contains some grammar mistakes and expressions that are not natural.

---

> ### Author Response · Authors · 2024-07-18
>
> **The proposed model is not entirely accurate. A constraint \mathbf{X} > \mathcal{0} should be imposed on the transportation matrix \mathbf{X}. Without this constraint, the conclusion that \nabla f is s-Lipschitz cannot be guaranteed, and the partition matrix \mathbf{G} cannot be obtained by assigning each node i to the (\arg\max_i \mathbf{x}_i)-th partition.**
>
> Thank you for spotting this. It is true that a nonnegativity constraint is necessary. We implicitly considered it since it is present in the definition of the transportation polytope but forgot to write it in equation (12). We have added it in the revised version.
>
>
> **The selection of parameters in the proposed model significantly influences its effectiveness. While size parameters can be set using domain knowledge, this requires strong prior knowledge of the dataset's size distribution, which is impractical in real-world unsupervised applications. If these size parameters are determined using other algorithms, the performance of the proposed model will be heavily influenced by those algorithms. Thus, the question arises: how can size parameters be determined in a reasonable manner, and what method was used in this paper? Additionally, the regularization parameter \lambda significantly affects the sparsity of the transportation matrix \mathbf{X}. Specifically, a smaller $\lambda$ results in a sparser \mathbf{X}. Why is \lambda set to 1 when \alpha is set to 1/2? If \lambda is set to 1, can it be guaranteed that the nonzero entries of the matrix \mathbf{X} be n + k - 1?**
>
> The size parameters can either be set using prior domain knowledge. This is the additional benefit of our algorithm when compared to an algorithm like spectral clustering for example. If no such prior knowledge is present, the most straightforward way to set them would be to use the idea used in spectral clustering: weighting the nodes using their degrees and look for clusters (set of nodes) that weigh roughly the same.
>
> A smaller $lamba$ does not guarantee more sparsity. We set the lambda in such a way as to find solutions that are vertices of the transportation polytope. The set of vertices of the transportations polytop is guaranteed to have a fixed number of nonzero elements $\in [n, n+k-1]$. A proof is available following theorem 3.4 in [4].
>
> **The theoretical proofs are neither complete nor entirely correct. In the proof of Proposition 1, the constrained problem (17) is transformed into the regularized problem (18). A key issue is whether this transformation is indeed equivalent. If the constraints in problem (17) are moved to the objective as a regularization, a regularization parameter must be added, which will determine the equivalence relationship between problems (17) and (18). The proof of Proposition 1 seems to suggest that, since problem (17) can be transformed into the form of classical optimal transport, the n + k - 1 nonzero entries of matrix \mathbf{X} can be guaranteed. This is not entirely accurate. The sparsity of matrix \mathbf{X} is also influenced by the selection of the parameter \lambda. Moreover, in Proposition 2, it should be stated that f + g is convex rather than concave, and the proof is notably incomplete. It is not straightforward to conclude that the nonconvex proximal GD algorithm can globally converge in a special case based solely on the literature by Li & Lin (2015). Detailed supplementary proof is necessary.**
>
> Problems (17) and (18) are equivalent and do not warrant a regularization parameter because the indicator function is either 0 or positive infinity. Any regularization would not modify that value. Please refer to slide 17 in: https://www.stat.cmu.edu/~ryantibs/convexopt/lectures/prox-grad.pdf.
>
> $f+g$ is never stated to be concave nor convex. It is only stated to be coercive. This is the same assumption used in [2].
>
> **The optimization algorithm is incomplete. The optimization procedure provided in the proof of Proposition 1 lacks sufficient detail for readers to fully understand it. Furthermore, the optimization procedure does not match the steps listed in Algorithm 1. Additionally, there is no explanation on how to initialize the partition matrix \mathbf{G}.**
>
> It is unclear to us how the optimization procedure does not match the steps listed in the algorithm. A thorough explanation of the algorithm can be found in the original paper [2]. Following standard practice, the matrix $\mathbf{G}$ can either be initialized randomly or using a partition of the data obtained by some other algorithm.

---

> ### Author Response · Authors · 2024-07-18
>
> (continued)
>
> **The literature review is incomplete. For balanced clustering, constrained clustering, and Gromov-Wasserstein graph clustering, numerous related and recent approaches exist. For instance, the following literature should be reviewed for balanced clustering**
>
> We have added the relevant references to the revised version. Note that [9] was already used as a baseline in the original version. [1] was not added due to being mostly about graph embedding instead of clustering.
>
> **The experiments are neither complete nor convincing enough. The experimental settings are not described in sufficient detail to reproduce the reported results. It is recommended that the proposed approach be compared with more related works. The clustering performance is evaluated using only a single metric, ARI, and its value range is [-1, 1] rather than [-0.5, 1]. Additionally, the convergence results shown in Figure 1 clearly indicate that the proposed algorithm fails to converge. Why can't the convergence of the proposed algorithm be guaranteed?**
>
>
> To ensure reproducibility, we plan to release the code upon acceptance of the paper. In the revised version, we have included an anonymized link to our source code and specified the implementations used for the baselines.
>
> The implementation of the ARI available states that the output values lie in [-0.5, 1]. It states that the output value is a “Similarity score between -0.5 and 1.0. Random labelings have an ARI close to 0.0. 1.0 stands for perfect match.” Please refer to: https://scikit-learn.org/stable/modules/generated/sklearn.metrics.adjusted_rand_score.html
>
> The papers for the baselines [1] and [3] used AMI as their metric. Since most datasets we used were balanced, we believe that ARI is a better choice.
>
> We have added the baselines used in [1]. We have the following observations to make:
> We reported results on the naturally occuring graphs only due to excessive run times over the image graphs.
> Louvain and infomap do not allow to specify the number of clusters. Comparison between partitions with a different number of clusters using ARI is not meaningful. As a result, we only reported results for datasets in which these algorithms consistently recovered the ground-truth number of clusters. We excluded Louvain from our analysis since it failed to find the ground-truth number of clusters in any of the runs.
> We also excluded Fluid from our analysis since it requires graphs with a single connected component, which was not the case for any of the naturally occurring graphs we studied.
> Additionally, as described in the paper, the convergence of our algorithm is guaranteed for a small $alpha$. The convergence plots show behavior for $\alpha=1/2$ which was the parameter used for all the experiments. It was selected as a default value to show that it is not necessary to compute the Lipschitz constant to obtain satisfactory performance in practice (even if it means theoretical convergence is not guaranteed).
>
> **There are many unclear descriptions in this paper. All abbreviations should be accompanied by their full terms upon first use. Additionally, why is the proximal GD algorithm referred to as the accelerated proximal GD algorithm? What aspect reflects the acceleration? There are many variables that are not mentioned in the notation section. In formula (9), what does “<>” mean? Several relevant references are not cited, such as those for discrete optimal transport, the Kantorovich formulation, and the network simplex algorithm. The statements "the proposed algorithm has globally convergence guarantees" in the abstract and conclusion are incorrect, as this is only valid in a special case, and the proof is incomplete.**
>
> We tried to make notations clearer in the revised version. A table of notations was added in the appendix. <> is the standard notation for the Frobenius inner product)
>
> We added the references that were asked for in the revised version.
>
> The term "global convergence" refers to the property of an algorithm where the generated points converge to a critical point, regardless of the initial starting point. However, it is important to note that the definition of global convergence can vary depending on the context. For an insightful discussion on the topic, please refer to this link:
> https://math.stackexchange.com/questions/905327/definition-of-global-convergence.

---

> ### Author Response · Authors · 2024-07-18
>
> (continued)
>
> **There are numerous typos in this paper. In formula (6), "W" should be in bold. Below formula (9), "c_{ij}" should be "m_{ij}". In formula (12), below the notation "\min", the optimization variable \mathbf{X} should be indicated. In the proof of Proposition 2, "coercive" should be "concave".**
>
> Thank you for spotting these typos. We have corrected them in the revised version. Please, note that the word coercive is not a typo.
>
>
> **The overall language expression in this article requires further revision. The current version contains some grammar mistakes and expressions that are not natural.**
>
> Since no expressions have been listed by the reviewer we tried to make the writing clearer by taking into account the explicit modifications that were asked for by the other reviewers.
>
> [1] Chowdhury, S., & Needham, T. (2021, March). Generalized spectral clustering via Gromov-Wasserstein learning. In International Conference on Artificial Intelligence and Statistics (pp. 712-720). PMLR.
>
> [2] Li, H., & Lin, Z. (2015). Accelerated proximal gradient methods for nonconvex programming. Advances in neural information processing systems, 28.
>
> [3] Xu, H., Luo, D., & Carin, L. (2019). Scalable Gromov-Wasserstein learning for graph partitioning and matching. Advances in neural information processing systems, 32.
>
> [4] Peyré, G., & Cuturi, M. (2019). Computational optimal transport: With applications to data science. Foundations and Trends® in Machine Learning, 11(5-6), 355-607.

---

### Review · Reviewer_RrfN · 2024-06-19

**Summary Of Contributions:**

This paper proposes a new graph cut algorithm for partitioning graphs under arbitrary size constraints. The graph cut problem is formulated as a Gromov-Wasserstein optimal transport problem with a concave regularizer. An accelerated proximal gradient descent algorithm is proposed to solve the problem, which has global convergence guarantees. Experiments on real-world graphs and graphs built from image datasets demonstrate the effectiveness of the proposed method.

**Audience:**

Yes

**Broader Impact Concerns:**

N.A.

**Claims And Evidence:**

Yes

**Requested Changes:**

1. To better understand the scalability of the proposed method, it would be informative to include an analysis of how the empirical runtime varies with graph size and the number of clusters. The current runtime plot only compares the method to spectral clustering.

2. While the experiments focus on comparisons with spectral clustering and other OT-based methods, the paper would be strengthened by incorporating comparisons to additional state-of-the-art graph clustering techniques beyond those based on optimal transport. This would provide a more comprehensive evaluation of the proposed method's performance.

3. The notation used throughout the paper is dense, and some terms are introduced without sufficient explanation. Including a table that summarizes the notation and symbols used would greatly assist readers in following the mathematical formulations more easily, enhancing the overall readability of the paper.

**Strengths And Weaknesses:**

Strengths

1. The paper presents a novel graph cut algorithm that leverages optimal transport to partition graphs with flexible size constraints. This approach extends the capabilities of popular normalized and ratio cut methods, enabling the incorporation of desired cluster size distributions.

2. The motivation for the proposed method is well-articulated, highlighting the limitations of existing normalized/ratio cut techniques. The paper demonstrates promising results on imbalanced datasets, which are prevalent in real-world scenarios.

3. The theoretical analysis provided in the paper is comprehensive, offering proofs for the sparsity of solutions, algorithm convergence, and computational complexity. Notably, the additional computational cost compared to spectral clustering is limited to an O(log n) factor.


Weaknesses

1. Although the paper suggests setting size parameters using domain knowledge, random guesses, or cluster size distributions from other algorithms, further discussion or experiments investigating the sensitivity of results to these size parameters would be valuable. Providing guidance on best practices for determining size parameters in the absence of prior knowledge would enhance the paper's practical utility.

2. The sensitivity of the proposed method to hyperparameters is acknowledged but not extensively investigated. Including a detailed sensitivity analysis that explores the impact of different hyperparameter settings on performance would offer valuable insights to users when selecting appropriate parameters for various datasets.

3. The paper would benefit from ablation studies that elucidate the contribution of different components within the proposed method, such as the concave regularizer. These studies would provide a clearer understanding of the role and importance of each component.

4. The results presented in Table 5 appear to exceed the valid range of the Adjusted Rand Index (ARI), which typically falls between -0.5 and 1. Clarification or correction of these results is necessary to ensure accuracy and consistency.

---

> ### Author Response · Authors · 2024-07-18
>
> **To better understand the scalability of the proposed method, it would be informative to include an analysis of how the empirical runtime varies with graph size and the number of clusters. The current runtime plot only compares the method to spectral clustering.**
>
> We added a plot for runtimes as a function of the number of clusters in the revised version. Spectral clustering was the only comparative method because our algorithm optimizes the same criterion and we tried to show that our algorithm can serve as a more efficient alternative when there is a large number of nodes.
>
> **While the experiments focus on comparisons with spectral clustering and other OT-based methods, the paper would be strengthened by incorporating comparisons to additional state-of-the-art graph clustering techniques beyond those based on optimal transport. This would provide a more comprehensive evaluation of the proposed method's performance.**
>
> We have added the baselines used in [1]. We have the following comments to make:
> - We reported results on the naturally occuring graphs only due to excessive run times over the image graphs.
> - Louvain and infomap do not allow to specify the number of clusters. Comparison between partitions with a different number of clusters using ARI is not meaningful. As a result, we only reported results for datasets in which these algorithms consistently recovered the ground-truth number of clusters.
> - We excluded Louvain from our analysis since it failed to find the ground-truth number of clusters in any of the runs.
> We also excluded Fluid from our analysis since it requires graphs with a single connected component, which was not the case for any of the naturally occurring graphs we studied.
>
> **The notation used throughout the paper is dense, and some terms are introduced without sufficient explanation. Including a table that summarizes the notation and symbols used would greatly assist readers in following the mathematical formulations more easily, enhancing the overall readability of the paper.**
>
> We added a table of symbols to complement the list of notations introduced in the original version. It can be found in the appendix.

---

### Review · Reviewer_wByg · 2024-07-08

**Summary Of Contributions:**

This submission studies a method of partitioning graphs via optimal transport. To illustrate their method, consider the following graph clustering problem, sometimes called $k$-balanced partitioning [1]. Given a weighted undirected graph $G = (V, E, w^G)$, and an integer $k > 0$, output a partition of $V$ into $k$ cuts $S\_1, \ldots, S\_k \subseteq V$ where simultaneously (1) the number of vertices in each partition is identical, and (2) the total capacity of edges across the partitions is minimized.

Finding optimal solutions for minimum $k$-way partitioning is significantly consequential from both a practical and theoretical perspective. Practically, the problem is used to model tasks in image segmentation, VLSI design, parallel computing, and many other domains. Theoretically, the problem is famously intractable; it is known to be NP-hard, and there has been extensive theory developed to understand what algorithmic techniques lead to provably efficient / accurate methods [2][3].

The authors propose relaxing $k$-balanced partitioning to the following non-convex program. Let $\mathbf{L} \in \mathbb{R}^{n \times n}$ denote the Laplacian matrix of $G$, where $n$ is the number of vertices, and $\langle \cdot, \cdot \rangle$ denote the Froebenius inner product.
$$
\begin{aligned}
& \min
& & \langle \mathbf{X} \mathbf{X}^{\top}, \mathbf{L} \rangle \\\\
& \textup{s.t.}
& & \mathbf{X} \mathbf{1} = \tfrac{1}{n} \cdot \mathbf{1} \\\\
& & & \mathbf{X}^{\top} \mathbf{1} = \tfrac{1}{k} \cdot \mathbf{1} \\\\
& & & \mathbf{X} \in \mathbb{R}\_{\geq 0}^{n \times k}
\end{aligned}
$$

One can see this is a relaxation by considering an integral solution which assigns $\mathbf{X}\_{ij} = \frac{1}{n}$ if and only if $i \in S\_j$. The authors interpret solving this program as solving an optimal transport (Gromov-Wasserstein) problem. Fix an initial distribution $\mu = \frac{1}{n} \cdot \mathbf{1}\_{n}$ uniform over the vertices, and terminal distribution $\nu = \frac{1}{k} \cdot \mathbf{1}\_{k}$ uniform $k$ disjoint elements, then computing optimal $\mathbf{X}$ is equivalent to finding the minimum cost transportation plan $\mathbf{X}$ where $\mathbf{X}\_{ij}$ denotes how much mass in $\nu\_j$ should comprise of mass in $\mu\_i$.

The core contributions of this work are then the following.

- The authors define the above non-convex relaxation for $k$-balanced partitioning, as well as other balanced partitioning problems. For example, they consider the task of $k$-balanced partitioning where cuts must now have identical volume instead of just the same number of vertices. The non-convex programs differ in how the initial, and terminal distributions $\mu$ and $\nu$ are chosen. The authors observe that freedom in choosing $\mu$ and $\nu$ allow for tackling a wide variety of graph partitioning tasks.

- The authors then impose a matrix-norm regularizer to the above program, and provide a proximal-point method for solving the resulting regularized problem. Leveraging a framework of [4], the authors prove that their method converges to a critical point given an appropriate choice of step size.

- Finally, the authors provide a promising set of experimental results supporting the viability of their method.

[1] Andreev, K. and Räcke, H., 2004, "Balanced graph partitioning."

[2] Krauthgamer, R., Naor, J. and Schwartz, R., 2009. "Partitioning graphs into balanced components."

[3] Bansal, N., Feige, U., Krauthgamer, R., Makarychev, K., Nagarajan, V., Seffi, J. and Schwartz, R., 2014. "Min-max graph partitioning and small set expansion."

[4] Li, H. and Lin, Z., 2015. "Accelerated proximal gradient methods for nonconvex programming."

**Audience:**

Yes

**Claims And Evidence:**

No

**Requested Changes:**

- [critical] It would be helpful to add some commentary regarding the references listed above, in particular those pertaining to modeling graph partitioning as a Gromov-Wasserstein problem. Alternatively, please explain why the references are not relevant to the current work.

- [critical] It would be helpful to clarify the writing style by addressing the above questions in the body of the paper.

- [non-critical] For the experiments demonstrating that the output cluster size distribution matches the terminal distribution given to the non-convex program, it would be helpful to test the method on a toy dataset generated by a stochastic block model where the block sizes are arbitrarily chosen.

- [nit] There are a couple small typos in the paper.
  - "where an self-representation matrix" -> "where a..." (page 1)
  - "there some weaknesses" -> "there are..." (page 1)
  - "this means that its adjacency matrix is diagonal" -> "this means that the adjacency matrix of $\mathcal{G}_{dc}$ is diagonal" (page 2)
  - "with an weighted" -> "with a..." (page 3)
  - "identity matrix$\mathbf{I}$ -> missing space (page 5)
  - "according to Li & Lin..." -> it is helpful to cite which theorem is being used (page 7)
  - "using ground truth..." -> capitalize "u" (page 10)
  - "and this for any..." -> "and thus (?)" (page 11)

**Strengths And Weaknesses:**

STRENGTHS
- The experimental results are promising.

In one set of experiments, the authors compare their methods (OT-rcut, and OT-ncut) to spectral clustering, and current state of the art optimal transport-based methods (S-GWL [5], and SpecGWL [6]). They apply the methods to two classes of data: (1) datasets where graph structure is inherently present (the DBLP citation network, ACM co-author citation network, an emailing network between large European research institutions, and an interaction network between villages in India), and (2) datasets where graph structure must be constructed (MNIST, Fashion-MNIST, KMNIST). For the latter category, the authors apply different subspace clustering techniques to build network representations. The authors report on running time, and ability to recover ground truth clusters as quantified by the Adjusted Rand Index. Across almost all tasks, their methods demonstrate an advantage in terms of running time, and ability to recover ground truth.


WEAKNESSES
- The work lacks some amount of literature review, making it more difficult to understand its novelty.

Section 5 of [7] seems to apply optimal transport to partition graphs in a similar manner. There is also extensive literature related to approximating the Gromov-Wasserstein distance for various choice of cost functionals [8][9]. Additionally, there is extensive literature on using proximal point methods and other numerical methods to solve optimal transport problems when the cost functional is linear [10][11] (and references therein). Though the present submission concerns solving a non-convex optimal transport problem, their method only converges to a critical point. In designing algorithms which converge to a critical point, related algorithms which work for convex settings are typically relevant.

- The writing style can be greatly improved. Technical notation is difficult to parse at times. When the work first defines the Gromov-Wasserstein distance, its use of $\otimes$ to denote a tensor-matrix product does not make it immediately clear which choice of indices the product is contracting on.

Seemingly important technical details are omitted. An important point is made regarding the use of a concave regularizer to enforce sparsity. What is the matrix norm being used to regularize the Gromov-Wasserstein problem in equation 17? Description of the results are a bit imprecise. What is meant when it is said that the proximal GD algorithm "has global guarantees"? Convergence results present in [4] are only to critical points in the objective.

There are inconsistencies present in how experimental data is reported. What are the units when running time is reported? It is also noted that ARI "takes values between 1 and -0.5" (page 9). However, the ARI scores reported for the imbalanced dataset experiments (table 5) range much larger than 1. How are these values scaled?

[4] Li, H. and Lin, Z., 2015. "Accelerated proximal gradient methods for nonconvex programming."

[5] Xu, H., Luo, D. and Carin, L., 2019. "Scalable Gromov-Wasserstein learning for graph partitioning and matching."

[6] Chowdhury, S. and Needham, T., 2021, "Generalized spectral clustering via Gromov-Wasserstein learning."

[7] Abrishami, T., Guillen, N., Rule, P., Schutzman, Z., Solomon, J., Weighill, T. and Wu, S., 2020. "Geometry of graph partitions via optimal transport."

[8] Altschuler, J., Bach, F., Rudi, A. and Weed, J., 2018. "Approximating the quadratic transportation metric in near-linear time."

[9] Quanrud, K., 2018. "Approximating optimal transport with linear programs."

[10] Jambulapati, A., Sidford, A. and Tian, K., 2019. "A direct $\tilde{O}(1/\epsilon)$ iteration parallel algorithm for optimal transport."

[11] Altschuler, J., Niles-Weed, J. and Rigollet, P., 2017. "Near-linear time approximation algorithms for optimal transport via Sinkhorn iteration."

---

> ### Author Response · Authors · 2024-07-18
>
> **Description of the results are a bit imprecise. What is meant when it is said that the proximal GD algorithm "has global guarantees"? Convergence results present in [4] are only to critical points in the objective.**
>
>  The term "global convergence" refers to the property of an algorithm where the generated points converge to a critical point, regardless of the initial starting point. However, it is important to note that the definition of global convergence can vary depending on the context. For an insightful discussion on the topic, please refer to this link:
> https://math.stackexchange.com/questions/905327/definition-of-global-convergence.
>
> **What is the matrix norm being used to regularize the Gromov-Wasserstein problem in equation 17**
>
>  ||.|| is the Frobenius norm. We have clarified this in the revised version.
>
> **What are the units when running time is reported? It is also noted that ARI "takes values between 1 and -0.5" (page 9). However, the ARI scores reported for the imbalanced dataset experiments (table 5) range much larger than 1. How are these values scaled?**
>
>  Thank you for noticing that, we have multiplied results by 100 for readability but omitted to mention it. We have restored those values to their original scale to avoid confusion.
>
> **The writing style can be greatly improved. Technical notation is difficult to parse at times. When the work first defines the Gromov-Wasserstein distance, its use of (x) to denote a tensor-matrix product does not make it immediately clear which choice of indices the product is contracting on.**
>
>  The notation for the tensor-matrix product was directly taken from [1]. An entry based notation was added to the revised version
>
>
> **It would be helpful to add some commentary regarding the references listed above, in particular those pertaining to modeling graph partitioning as a Gromov-Wasserstein problem. Alternatively, please explain why the references are not relevant to the current work.**
>
>  Thank you for your suggestions, we have added the relevant literature you cited in the revised version.
>
> **Typos**
>
>  Thank you for spotting the listed typos. We have corrected them in the revised version.
>
> [1] Peyré, G., Cuturi, M., & Solomon, J. (2016, June). Gromov-wasserstein averaging of kernel and distance matrices. In International conference on machine learning (pp. 2664-2672). PMLR.

---

### Author Response · Authors · 2024-07-18

We wish to express our gratitude to the reviewers for their thorough reviews. Their feedback is greatly appreciated and we believe the paper has improved thanks to it.

---

### Decision · Action_Editor_acxj · 2024-09-12

**Recommendation:** Accept with minor revision

**Comment:**

There were significance differences among the reviewers about how new, accurate and convincing this paper is.  I believe that, after the revisions made during rebuttal, the paper can be accepted. I will however ask for a few further revisions:

- There was significant confusion among the reviewers regarding the claims of global convergence. I understand the authors' assigned meaning, but I think it would really clarify the contribution of the paper to include "global convergence to a critical point" or "to a stationary point" wherever this is mentioned.

- The authors should include a section on limitations of their method, which should include a discussion of the need for prior domain knowledge and how this can ameliorated in large applications.

**Audience:**

Yes. The area of graph partitioning via optimal transport method is of current interest.

**Claims And Evidence:**

Yes, after the revisions made during the review process.